# Rapid and Broad Immune Efficacy of a Recombinant Five-Antigen Vaccine against *Staphylococcus aureus* Infection in Animal Models

**DOI:** 10.3390/vaccines8010134

**Published:** 2020-03-18

**Authors:** Hao Zeng, Feng Yang, Qiang Feng, Jinyong Zhang, Jiang Gu, Haiming Jing, Changzhi Cai, Liming Xu, Xi Yang, Xin Xia, Ni Zeng, Shaowen Fan, Quanming Zou

**Affiliations:** 1National Engineering Research Center of Immunological Products, Department of Microbiology and Biochemical Pharmacy, College of Pharmacy, Third Military Medical University, Chongqing 400038, China; yang.feng@olymvax.com (F.Y.); zhangjy198217@126.com (J.Z.); jianggu2012@163.com (J.G.); jhm2020@aliyun.com (H.J.); 2Chengdu Olymvax Biotechnology Co., Ltd., Chengdu 611731, Sichuan, China; menglongcai@aliyun.com (C.C.); LMX0203@aliyun.com (L.X.); yangxi213@aliyun.com (X.Y.); xx558998@aliyun.com (X.X.); zengni_jennie@aliyun.com (N.Z.); fan.shaowen@olymvax.com (S.F.); 3Chongqing Collaborative Innovation Center for Functional Food, Chongqing University of Education, Chongqing 400065, China; fengqiang@cque.edu.cn

**Keywords:** *Staphylococcus aureus*, five-subunit vaccine, immune efficacy, animal models, perioperative period vaccination

## Abstract

*Staphylococcus aureus* (*S. aureus*) is a leading cause of both healthcare-and community-associated infections globally, which result in severe disease and readily developing antibiotic resistance. Developing an efficacious vaccine against *S. aureus* is urgently required. In the present study, we selected five conserved antigens, including the secreted factors α-hemolysin (Hla), staphylococcal enterotoxin B (SEB) and the three surface proteins staphylococcal protein A (SpA), iron surface determinant B N2 domain (IsdB-N2) and manganese transport protein C (MntC). They were all well-characterized virulence factor of *S. aureus* and developed a recombinant five-antigen *S. aureus* vaccine (rFSAV), rFSAV provided consistent protection in *S. aureus* lethal sepsis and pneumonia mouse models, and it showed broad immune protection when challenged with a panel of epidemiologically relevant *S. aureus* strains. Meanwhile, rFSAV immunized mice were able to induce comprehensive cellular and humoral immune responses to reduce bacterial loads, inflammatory cytokine expression, inflammatory cell infiltration and decrease pathology after challenge with a sub-lethal dose of *S. aureus.* Moreover, the importance of specific antibodies in protection was demonstrated by antibody function tests in vitro and in vivo. Altogether, our data demonstrate that rFSAV is a potentially promising vaccine candidate for defensing against *S. aureus* infection.

## 1. Introduction

*Staphylococcus aureus* (*S. aureus*) is an important pathogen that causes hospital and community infections, such as sepsis, pneumonia, infective endocarditis, septic arthritis, fracture fixation and invasive infections, and its mortality rate is up to 20% [1]. With the inappropriate and irrational use of antibiotics, cases of methicillin-resistant staphylococcus aureus (MRSA), which is characterized by its high pathogenicity, extensive spreading and outbreaks and multidrug resistance, have increased [2]. An estimate produced by the European Centres for Disease Control and Prevention (CDC) shows that 1.7 million healthcare-associated infections and 5400 infection-associated deaths are due to MRSA [3]. In addition, it is difficult to treat severe MRSA infections [4]. Therefore, *S. aureus* was declared Priority 2: HIGH by a global priority list of antibiotic-resistant bacteria [4].

The development of antibiotic resistance has created a global challenge for treating *S. aureus* infections [5] and the rate of bacterial resistance is increasing faster than the research and development of new antibiotics [4]. Safe and effective vaccines are urgently needed. Nine *S. aureus* vaccines from seven companies [6,7], including Merck [8,9], Nabi [10,11], Vaccine Research International Plc [12], Pfizer [13,14], Novartis [15], GSK [16] and NIAID [17], have conducted clinical research so far. What is more, Nabi’s StaphVax, Merck’s V710 and Pfizer’s SA4Ag vaccines have conducted efficacy studies [8,10,12]. Whereas several candidates have failed to show a protective efficacy in human subjects [5]. The main reasons for the failure of some *S. aureus* vaccine may include: (1) Several candidates are single components but *S. aureus* is phenotypic variability. *S. aureus* is able to switch off toxins, capsule and adhesions during different phases of growth, local environment and in response to host defences, including antibodies [18]. (2) Several candidates focus only on functional antibodies, however protective immunity from *S. aureus* infection is incompletely defined as opsonic or neutralizing antibody [18,19]. A robust level of *S. aureus* vaccine-induced antibodies may be important, but insufficient, for inducing protective efficacy [18].

The failure of the *S. aureus* vaccines has brought great challenges to vaccine research and development. We should adopt new strategies to study *S. aureus* vaccines. First, we designed a “cocktail” vaccine formulation for multiple targets. This five-antigen containing vaccine has more targets than other reported *S. aureus* vaccines. Our vaccine contains five antigen targets, including SpA, Hla, IsdB-N2, SEB and MntC. These antigens contain bacterial toxin molecules, membrane proteins and proteins closely associated with bacterial growth metabolism. The vaccine using these proteins as antigens offers enhanced protection by inhibiting or blocking key pathogenic links, such as bacterial adhesion, toxin release, metabolism and immune escape. Furthermore, through molecular mutation and fusion design, we removed the harmful toxic activity of these proteins and identified that these protein antigens could maintain a fair level of immunogenicity. Second, our vaccine works by multiple immunologic mechanisms, and induced robust antigen specific humoral and cellular immune response, obviously producing immuno-protection against different sources of *S. aureus* strain infections in animal models of systemic infection and pulmonary infection.

## 2. Materials and Methods

### 2.1. Ethics Statement

All animal care and use protocols in this study were performed in accordance with the Regulations for the Administration of Affairs Concerning Experimental Animals approved by the State Council of the People’s Republic of China. All animal experiments in this study were approved by the Animal Ethical and Experimental Committee of the Third Military Medical University (Chongqing, Permit No. 2011-04) in accordance with their rules and regulations. All surgical procedures were performed under sodium pentobarbital anaesthesia, and all efforts were made to minimize suffering.

### 2.2. Bacterial Strains and Culture Methods

The standard *S. aureus* strain MRSA252 was purchased from ATCC (Manassas, VA, USA). Clinical strains of 8 *S. aureus* isolates were collected from 6 hospitals in different districts of China (Appendix A). Bacterial strains were cultured in tryptic soy broth, and the cell concentration was determined spectrophotometrically at 600 nm (OD600).

### 2.3. Animals

BALB/c mice (females, 6–8 weeks, 16–18 g) and C57BL/6 mice (females, 6–8 weeks, 16–18 g) were purchased from Beijing HFK Bioscience Limited Company (Beijing, People’s Republic of China) and kept under specific pathogen-free (SPF) conditions. Female New Zealand white rabbits (weighing 2.00 ± 0.20 kg) were provided by TengXin Company (Chongqing, China).

### 2.4. Cloning, Expression and Purification of Recombinant Vaccine Antigens

Experimental methods of cloning, expression and purification of recombinant vaccine antigens are described in detail in Appendix A. The protein purity was determined by SDS-PAGE and high-performance liquid chromatography (HPLC). The protein concentration was determined using the bicinchoninic acid (BCA) method (Pierce). The endotoxin content was detected using a Tachypleus Amebocyte Lysate assay (Houshiji Cod Inc., Xiamen, China). The endotoxin levels were <2.5 pg/μg for all four recombinant proteins [20]. The four purified proteins in His buffer were formulated with AlPO_4_ (American General Chemistry, Fairfield, Connecticut, USA), and we named the recombinant five-antigen *S. aureus* vaccine as FSAV.

### 2.5. Mouse Immunization

For active immunization, mice were intramuscularly injected with 600 μL of recombinant five-antigen *S. aureus* vaccine (rFSAV) (30 μg mHI_N2_, MntC, mSEB and SpA5, respectively), His buffer plus AlPO_4_ adjuvant, or His buffer alone as the control on days 0, 3 and 7, and the mice were infected on day 18 (Appendix A).

For passive immunization, rFSAV polyclonal antibodies (rFSAV-pcAb) were generated in rabbits based on a previously published method [21]. Next, the IgG in the serum from immunized or unimmunized rabbits was purified by affinity chromatography with a protein A column (GE Healthcare, USA), and desalted with PBS. The rFSAV-pcAb or negative-pcAb concentrations were determined by the BCA method [22] and adjusted to a final concentration of 20 mg/mL. Two hours before infection, mice were injected intravenously with 2 mg of rFSAV-pcAb or 2 mg of negative-pcAb.

### 2.6. ELISA

Mouse serum samples were collected for an enzyme-linked immunosorbent assay (ELISA), which was performed as previously described [23]. In brief, microtiter plate wells (Corning Incorporated, New York City, NY, USA) were coated with mHI_N2_, MntC, mSEB or SpA5 (200 ng per well) in 0.05 M carbonate buffer (pH 9.5) overnight at 4 °C. The primary antibodies were diluted serum samples, and the secondary antibodies were HRP-conjugated goat anti-mouse IgG, anti-mouse IgG1, anti-mouse IgG2a or anti-mouse IgG2b (Sigma). The optical density was measured at 450 nm, and the titres were defined as the highest dilution that yielded an absorbance value of more than twice the value of the pre-immune serum.

### 2.7. S. aureus Sepsis Mouse Model

For survival analyses in the *S. aureus* sepsis model, immunized BALB/c mice were intravenously infected with *S. aureus* and monitored for survival for 10 days after infection [24,25]. The lethal doses of the *S. aureus* strains MRSA252, JN-75, CQ-19, GZ-02 and KM-22 were determined to be 6.0 × 10^8^, 3.0 × 10^8^, 5.0 × 10^8^, 7.0 × 10^8^ and 2.0 × 10^8^ colony-forming units (CFUs)/mouse, respectively. To measure bacterial burdens in the sepsis model mice, a sub-lethal dose (3.0 × 10^8^ CFUs/mouse) of MRSA252 was intravenously administered. The peripheral blood was collected in heparin anticoagulant tubes. In addition, kidney and spleens were removed, weighed and homogenized in 1 mL of PBS 1 or 3 days after infection. All samples were then plated on MHA plates at a 10-fold serial dilution and cultured at 37 °C overnight. The number of CFUs per gram of tissue (CFUs/g) was calculated from each plate.

### 2.8. S. aureus Pneumonia Mouse Model

For survival analyses in the mouse *S. aureus* acute pneumonia model, immunized C57BL/6 mice were anaesthetized with 30 mg/kg pentobarbital sodium followed by intratracheal injection with a lethal dose of *S. aureus* [26,27]; the number of deaths in each group was recorded every 12 h over a 7-day observation period post-challenge. The lethal doses of *S. aureus* strains MRSA252, BJ-04, CQ-SA77, GZ-19 and SJZ-23 were determined to be 1.0 × 10^8^, 9.0 × 10^8^, 4.0 × 10^8^, 3.0 × 10^8^ and 4.0 × 10^8^ CFUs/mouse, respectively. For bacterial burden, histopathology, inflammatory cell and cytokine analyses, mice were infected by intratracheal injection with 5.0 × 10^7^ CFUs/mouse of MRSA252. Then, the lung tissues were collected, weighed and homogenized in 1 mL of sterilized PBS buffer 24 h after infection to determine CFUs.

### 2.9. Histological Analysis

Lung samples collected from the pneumonia model mice were fixed in 10% neutral formalin, embedded in paraffin, sectioned and stained with haematoxylin and eosin (HE). A single pathologist viewed the sections at 100× magnification. Each lung section was given a score of 0–4 (no abnormality to most severe) according to established criteria based on hyperaemia, oedema, haemorrhaging and neutrophil infiltration [20].

### 2.10. Evaluation of Inflammation

To evaluate inflammation in the pneumonia model mice, neutrophil infiltration and proinflammatory responses were quantified. On the one hand, cells in bronchoalveolar lavage fluid (BALF) were collected from mice 24 h post-challenge and stained using the following antibodies: PE/Cy7 anti-mouse CD45 and APC/Cy7 anti-mouse Ly-6G (Biolegend, Inc., USA). Samples were then analysed using BD FACSArray software on a BD FACS Array flow cytometer (BD Biosciences). On the other hand, cytokines such as TNF-α, interleukin (IL)-1β and IL-6 in the BALF from mice were determined 24 h post-challenge using a Mouse Quantikine ELISA kit for TNF-α, IL-1β or IL-6 (R&D Systems, Minneapolis, Minnesota, USA), respectively, according to the manufacturer’s instructions.

### 2.11. Immune Response Assays

Immune response assays were performed to determine the cytokine production of splenocytes from immunized mice two weeks after the final immunization, as described previously [28]. In brief, spleens were removed, homogenized and suspended in 4 mL of PBS containing 2% foetal bovine serum (FBS, Gibco, Waltham, Massachusetts, USA). Next, the cells were treated with red blood cell (RBC) lysis buffer (ZSGB-BIO, China), washed, centrifuged and resuspended in RPMI 1640 (HyClone, USA) supplemented with 5% FBS and 4 mM L-glutamine and then adjusted to a concentration of 4 × 10^6^ cells/mL. The suspended cells were dispensed into 96-well culture plates (Corning, USA) at 100 μL/well and stimulated with 10 μg/well rFSAV antigens. The concentrations of interferon-γ (IFN-γ), IL-5 and IL-17A were measured in 72-h cultures by examining the cell-free culture supernatant using a Mouse Quantikine ELISA kit for IFN-γ, IL-5 or IL-17A (R&D Systems, Minneapolis, Minnesota, USA), respectively, according to the manufacturer’s instructions.

### 2.12. Opsonophagocytic Killing Assay

The antibody opsonophagocytic killing assay was carried out as described previously with appropriate modifications [29]. Briefly, the Human promyelocytic leukemia cells line (HL-60, ATCC: CCL-240) were differentiated into granulocyte-like cells in growth medium containing 0.8% N,N-dimethylformamide (Sigma-Aldrich, St Louis, MO, USA) for 4 days. The targeted strains were the 8 clinical isolates listed in Appendix A. rFSAV-pcAb and negative-pcAb were serially diluted (1:-1:128). The assay was performed in 96-well plates, with each well containing the following components: 40 μL of 4 × 10^5^ HL60 cells, 10^3^ CFU of targeted strains in 10 μL of opsonophagocytic buffer B (OBB), 20 μL of rFSAV-pcAb or negative-pcAb, and 10 μL infant rabbit serum as a complement source (Pel-Freez). After 95 min incubation, 10 μL of the reaction mixture from each well was spotted onto Mueller-Hinton Agar. CFUs were calculated by colony counter (Synbiosis, Cambridge, UK) after overnight incubation. Control samples were incubated with OBB instead of pcAb. The killing effect was defined as a reduction in CFUs compared with OBB control. The opsonophagocytic antibody level is expressed as an opsonization index equivalent to the dilution of serum that kills 50% of the bacteria using a linear interpolation algorithm estimated by using Opsotiter3 (Authorized by Dr. Moon H. Nahm, University of Alabama at Birmingham).

### 2.13. Hemolytic Activity Assay and Intestinal Toxin Activity Assay

Hemolytic activity assay and intestinal toxin activity assay were carried out based on a method established by us previously [29]. Briefly, for hemolytic activity assay, rabbit erythrocyte suspension in PBS (1%) was mixed with Hla, mHla and Hla pre-incubated with rFSAV-pcAb or negative-pcAb for 30 min. After a 30 min incubation at 37 °C, the mixtures were centrifuged, and then the hemolytic activity was determined by the release of hemoglobin, measured spectrophotometrically at 540 nm. For intestinal toxin activity assay, BALB/c mice were injected intraperitoneally with 20 mg of D-Galactosamine followed by an intramuscular injection of SEB, mSEB or SEB pre-incubated with rFSAV-pcAb or negative-pcAb for 30 min. Survival was monitored for 40 h after injection.

### 2.14. B Cell Apoptosis Assay

B cell apoptosis assay were performed to determine the neutralizing toxin effect of rFSAV-pcAb, as described previously [30]. Briefly, 100 μL of wild type SpA (SpA, 1.5 mg/mL) that had been incubated with 2 mg of rFSAV-pcAb or negative-pcAb 30 min prior to the start of the study was injected into the peritoneum of six-week-old female BALB/c mice. Spleens were removed and homogenized 4 h after injection. The cells were treated with red blood cell (RBC) lysis buffer and then white blood cells were stained with FITC anti-mouse CD3 and PE anti-mouse CD19 (Biolegend, Inc., San Diego, California, USA). Samples were then analysed using BD FACSArray software on a BD FACS Array flow cytometer (BD Biosciences).

### 2.15. Statistical Analysis

Data are presented as the mean ± standard deviation (SD) or mean ± standard error of the mean (SEM). Scoring experiments were performed in a blinded manner. Survival data were analysed using Kaplan-Meier survival curves. To calculate *p* values, a nonparametric Mann–Whitney test, log-rank test, Student’s *t*-test and one-way ANOVA with Bonferroni correction were used depending on the sample distribution and variation as mentioned in the figure legends. SPSS 26.0 (IBM, Inc., Armonk, New York, USA) and GraphPad Prism 8.0 (GraphPad Software, Inc., La Jolla, California, USA) were used to perform the statistical analyses. Significance was accepted at *p* < 0.05.

## 3. Results

### 3.1. Antigen Selection and Modification

rFSAV included five antigens: staphylococcal protein A (SpA), α-haemolysin (Hla), iron surface determinant B N2 domain (IsdB-N2), staphylococcal enterotoxin B (SEB) and manganese transport protein C (MntC). These antigens are surface and secreted factors previously shown to be protective and involved in *S. aureus* virulence.

The N-terminus of SpA contains an immunoglobulin binding domain that binds to mammalian IgG, and SpA is a superantigen that induces apoptosis of B cells [31]. Thus, it is a pluripotent virulence factor and was selected as a component of the *S. aureus* vaccine [30,32]. As shown in Figure 1A, the five active domains (E, D, A, B and C) of SpA are key for its superantigen activity and immune escape by *S. aureus*. There are two consecutive glutamines (QQ) at positions 7-8 and two consecutive aspartates (DD) at position 34-35 in domain E (Figure 1A, similar to sites in other domains), and these amino acids play a key biological role in SpA activity [30]. We modified the five active domains of SpA separately (E(KKAA), D(KKAA), A(KKAA), B(KKAA) and C(KKAA)) to form a non-toxic pentamer, named SpA5 (Figure 1A).

Hla, one of the best-characterized exotoxins of *S. aureus*, has strong haemolytic activity [26]. The histidine at position 35 plays an important role in the formation of the perforation complex [33,34]. We mutated this histidine to leucine (Leu) to remove the haemolytic activity of Hla (Figure 1B). IsdB is a highly conserved transmembrane protein encoded by the iron-regulated surface determinant system, containing 612 amino acids, which can be divided into two domains, N1 and N2, with different functions [35]. The N1 domain (91-239 amino acids) binds haemoglobin, the N2 domain (308-425 amino acids) binds heme and the function of N2 is based on the binding of N1 to haemoglobin [35]. In addition, we found that there is an important B cell epitope enrichment region by analysing the B cell epitope of N2. Therefore, we chose to connect domain N2 to Hla (H35L) with a linker, “GGGGS” (Figure 1B), and we named it mHI_N2_.

SEB, one of the most potent superantigens, is the main cause of human septic shock, the systemic inflammatory response and food poisoning and thus is classified as a class B biological warfare agent [36]. Amino acid mutations in the polar region of SEB (Y89 and Y94) and a single amino acid mutation (L45) in the hydrophobic domain can disable its ability to form a bridge between major histocompatibility complex class II molecules and maximally restore the conformation of SEB [37,38]. Therefore, we constructed a three-site (L45R, Y89A and Y94A) SEB mutant (Figure 1C) named mSEB.

MntC, an ATP-binding cassette (ABC) transporter system component, has been shown to play a role in manganese uptake [39]. The acquisition of manganese is important for *S. aureus* virulence and is associated with the survival and normal growth of *S. aureus* [40]. Additionally, MntC is a surface protein that is widely conserved in *S. aureus*, including MRSA and vancomycin-resistant *S. aureus* (VRSA) strains [41]. Therefore, we selected it as an antigen for rFSAV (Figure 1D).

As shown in Figure 1E, all 4 proteins (SpA5, mHI_N2_, mSEB and MntC) expressed in *E. coli* were soluble, and the purity after several rounds of chromatography was more than 95%, as determined by SDS-PAGE and high performance liquid chromatography (Appendix A). The molecular weights of these recombinant proteins were in accordance with their predicted molecular masses (32.0, 47.6, 28.3 and 32.3 kDa for SpA5, mHI_N2_, mSEB and MntC, respectively).

### 3.2. rFSAV Induces Consistent Immunoprotection against S. aureus in a Sepsis Model

First, mice were immunized with rFSAV and control combinations, and challenged with a lethal dose of *S. aureus* MRSA252 to investigate the protective effects of rFSAV. As shown in Table 1, at the end of the observation period, mice vaccinated with the bivalent combinations, trivalent combinations, and rFSAV exhibited higher mean survival rates (60%–87% survival) than did the AlPO_4_ alone group (25% survival) and the His buffer group (16% survival). Moreover, the mean survival rate of rFSAV group were the highest among all these 18 groups, up to 87%. In addition, the rFSAV group showed a greater protective effect than did vaccination with V710 which were shown in the available literature [24] (58% survival).

Further, the rFSAV-immunized mice were challenged with a sub-lethal dose of *S. aureus* MRSA252 to investigate the bacterial burden in the organs in a sepsis model. The results showed that the bacterial burden in the blood, kidneys and spleen was much lower in the rFSAV group than the AlPO_4_ control group at 1 day post-infection (*p*
_blood_ = 0.0033, *p*
_kidney_ = 0.0004 and *p*
_spleen_ = 0.0101, Figure 2A). Furthermore, the reduction in *S. aureus* CFUs at 3 days post-infection was enhanced in the rFSAV group compared to the AlPO_4_ group (*p*
_blood_ = 0.0004, *p*
_liver_ = 0.0001 and *p*
_spleen_ = 0.0001, Figure 2B). These results show that immunization with rFSAV protects mice against *S. aureus* infection by reducing the ability of the bacteria to colonize and directly attack organs, improving survival.

We next tested the protective efficacy of rFSAV in the sepsis model challenged at different time points. As shown in Figure 2C, immunized mice were challenged with a lethal dose of MRSA252 at 8, 18 and 68 days after the first immunization. At the end of the observation period, the survival rates of group D8, group D18 and group D68 were significantly higher than those of the adjuvant control groups (*p*
_D8_ = 0.0010, *p*
_D18_ < 0.0001 and *p*
_D68_ = 0.0040, respectively; Figure 2C). There was no significant difference in protection rates among the three groups (Figure 2C). These results indicate that rFSAV works quickly, and its protective effects last for at least 2 months.

To determine whether rFSAV provides broad protection in the *S. aureus* sepsis model, immunized mice were challenged with four different clinical strains of *S. aureus.* The four clinical strains were chosen from a library established in our lab based on their representation and diversity. The library contains more than 400 clinical isolates collected from different districts in China. Information regarding the 4 isolates is listed in Appendix A. As shown in Figure 2D–G, all 4 clinical strains exhibited different levels of pathogenicity and virulence in mice, as indicated by the survival rates in the His control groups. Compared with the AlPO_4_ controls, rFSAV protected 70% to 90% of mice from the clinical isolate challenge (*p*
_JN-75_ = 0.0007, *p*
_CQ-19_ < 0.0001, *p*
_GZ-02_ = 0.0025, *p*
_KM-22_ = 0.0018, Figure 2D–G).

### 3.3. rFSAV Vaccination Protects Mice from Pneumonia by Reducing Local Bacterial Burden and Inflammation

In the *S. aureus* pneumonia model, rFSAV also showed strong protective efficacy. C57BL/6 mice were immunized with rFSAV at days 0, 3 and 7 and then intratracheally infected with a lethal dose of MRSA252. As shown in Figure 3A, mice in the rFSAV immunization group exhibited survival rates (80%) that were significantly higher than those of mice in the His buffer control (*p* < 0.0001) and the AlPO_4_ adjuvant control (*p* = 0.0011) groups.

Next, the rFSAV-immunized mice were challenged with a sub-lethal dose of MRSA252 to investigate the protection mechanism. First, the lungs from immunized and control mice were harvested 24 h post-infection. Compared to the AlPO_4_ control mice, the immunized mice exhibited decreased pulmonary oedema post-infection, as measured by lung weight (*p* = 0.0054, Figure 3B). Furthermore, the rFSAV group showed significantly lower bacterial loads than the AlPO_4_ group at 24 h post-infection (*p* = 0.0003, Figure 3C). Second, the histological analysis showed that compared with those from AlPO_4_ control mice, the lungs from rFSAV-immunized mice exhibited reduced alveolar disruption, vascular leakage and deposition of bacterial microcolonies in the alveoli after infection (Figure 3D). In addition, mice immunized with rFSAV exhibited less inflammatory cell infiltration, bleeding and tissue damage than AlPO_4_ (Figure 3D,E). Third, markers of inflammation, including neutrophil infiltration and the production of proinflammatory cytokines, such as TNF-α, IL-1β and IL-6, production in the BALF were determined 24 h post-infection. As shown in Figure 3F–J, neutrophil number, percentage relative to other leukocytes and proinflammatory cytokine secretion were significantly reduced in the BALF of mice immunized with rFSAV, which was consistent with the lung histopathology results described above.

Furthermore, four clinical strains (Appendix A) were chosen from a library to determine whether rFSAV provides broad protection in the *S. aureus* pneumonia model. All 4 clinical strains exhibited different levels of pathogenicity and virulence in mice, as indicated by the survival rates in the His control groups. Compared with the AlPO_4_ control, rFSAV protected 70% to 80% of mice (*p*
_BJ-04_ = 0.0013, *p*
_CQ-SA77_ = 0.0002, *p*
_GZ-19_ = 0.0002, *p*
_SJZ-23_ = 0.0003, Figure 3K–N).

Taken together, these results confirm the broad protective efficacy of rFSAV in the *S. aureus* pneumonia model, which is attributable to reduced pulmonary oedema, bacterial burden, pathology and proinflammatory cytokine production.

### 3.4. rFSAV Vaccination Elicits Specific CD4 T-Cell Responses and a Rapid Humoral Immune Response

To analyse CD4 T-cell responses to the combined antigens induced by rFSAV, we performed two types of experiments. First, we examined the cytokine levels in the supernatants after stimulation with recombinant proteins. Compared with those from the AlPO_4_ control mice, the splenocytes from the vaccinated mice produced significantly more IFN-γ (*p* = 0.0054), IL-5 (*p* = 0.0005) and IL-17A (*p* = 0.0007) in response to stimulation with the recombinant proteins (Figure 4A–C). Second, we also measured the isotype profile (IgG1, IgG2a and IgG2b) of the antigen-specific IgGs induced by rFSAV antigens. As shown in Figure 4D, mHI_N2_, mSEB, MntC and SpA5 all induced high levels of antigen-specific IgG1, IgG2a and IgG2b subtypes. Furthermore, the level of antigen-specific IgG1 was higher than that of IgG2a (*p* < 0.0001). IgG1 and IgG2a are markers of Th2 and Th1 responses, respectively. These results suggest that rFSAV induced a Th2-biased response, which is consistent with the trend of cytokine production in the supernatants.

To further analyse antibody response time course, the specific IgG of each antigen in mice immunized with rFSAV was detected from days 0 to 42. As shown in Figure 4E, the level of antigen-specific IgGs was significantly higher on day 7 than day 0, and the mean fold increase (MFI) of mHI_N2_- and MntC-specific IgGs was greater than 4, which represents a positive conversion of antibody. Furthermore, the MFI of each antigen-specific IgG reached a peak on day 14. The MFI of SpA5-specific IgGs was more than 20, which was the lowest, and that of MntC-specific IgGs was the highest at greater than 2000 (Figure 4E).

### 3.5. rFSAV-pcAb Is Effectivity on Promoting Opsonophagocytosis and Neutralizing Toxins

To investigate the efficacy of the rFSAV-specific antibodies in vitro, we first performed an opsonophagocytic killing assay. As shown in Figure 5A, in the presence of HL60 phagocytic cells and complement, the mean opsonization index of rFSAV-pcAb against MRSA252 and 8 clinical *S. aureus* strains were from 41.8 to 126.6. Whereas, undiluted negative-pcAb cannot kill more than 50% of the bacteria, so the mean opsonization index of negative -pcAb against MRSA252 and 8 clinical *S. aureus* strains were all 2. These results indicate that rFSAV-pcAbs specifically increase opsonophagocytic activity of innate immune cells and killing of different clinical *S. aureus* strains.

We then examined the inhibitory efficacy of rFSAV-pcAb on *S. aureus* toxins including Hla, SpA and SEB. The hemolytic activity of Hla was detected by measuring the supernatant of 1% rabbit erythrocytes at 540 nm, which were incubated with Hla. As shown in Figure 5B, incubation of rFSAV-pcAb with Hla resulted in a significant inhibition of hemolytic activity compared with negative-pcAb group (*p* < 0.0001). Further, no significant differences in OD540 were observed when Hla was incubated with rFSAV-pcAb as compared to the His buffer and mHla control groups, indicating that rFSAV-pcAb is able to completely inhibit the hemolytic activity of Hla.

The B cell superantigen activity of SpA was assayed by measuring CD3^−^ CD19^+^ B cells in splenic tissue of mice treated with SpA. As shown in Figure 5C, the percentages of CD3^−^ CD19^+^ B cells of SpA group were significantly reduced than that of His buffer control group (*p* = 0.0001), which exhibited the potent B cell superantigen activity of SpA. However, after SpA was pre-incubated with rFSAV-pcAb, there were no significant difference in the percentages of CD3^−^ CD19^+^ B cells between rFSAV-pcAb group and His buffer control group (Figure 5C). The results show that rFSAV-pcAb is also able to inhibit the B cell superantigen activity of SpA.

The intestinal toxin activity of SEB can be evaluated by challenge test. We evaluated the impact of rFSAV-pcAb on suppressing intestinal toxin activity in wild type SEB. As shown in Figure 5D, after sensitizing with D-Galactosamine, all mice that were challenged with wild type SEB (25 μg) were dead within 24 h. In contrast, mice in the rFSAV-pcAb incubation group exhibited survival rates (90%) that were significantly higher than those of mice in the negative-pcAb incubation control group (*p* < 0.0001) and SEB positive control group (*p* < 0.0001, Figure 5D). These results confirm that rFSAV-pcAb has an inhibitory effect on the toxin activity of Hla, SpA and SEB.

### 3.6. Passive Immunization Protects Mice against Clinical S. aureus Strains in Murine Sepsis and Pneumonia Models

We first evaluated the protective efficacy of rFSAV-pcAbs against MRSA252 in vivo, and found that pcAb group displayed higher survival rates both in the murine sepsis model and the murine pneumonia model as compared to the negative-pcAb control group (*p* < 0.0001, Appendix A). Further, we examined the protective efficacy of passive immunization with rFSAV-pcAbs against clinical *S. aureus* strains. In a murine sepsis model, passively immunized mice were challenged with four different *S. aureus* clinical strains, JN-75, GZ-02, CQ-19 and KM-22. As shown in Figure 6A–D, compared to the negative-pcAb controls, rFSAV-pcAbs protected 70% to 90% of mice from the clinical isolate challenge (*p*
_JN-75_ = 0.0003, *p*
_GZ-02_ = 0.0002, *p*
_CQ-19_ < 0.0001, *p*
_KM-22_ < 0.0001). In the murine pneumonia model, we obtained similar results. Compared to the negative-pcAb controls, rFSAV-pcAbs protected 70% to 90% of mice from the clinical isolate challenge (*p*
_BJ-04_ < 0.0001, *p*
_CQ-SA77_ = 0.0003, *p*
_GZ-19_ = 0.0009, *p*
_SJZ-23_ = 0.0017, Figure 6E–H). These results strongly indicate that rFSAV-pcAbs can provide broad protection to mice challenged with different clinical strains of *S. aureus*.

## 4. Discussion

Over the past two decades, there have been numerous attempts to develop an effective *S. aureus* vaccine, with limited success to date. The development of such a vaccine faces a number of significant challenges, which have been extensively discussed elsewhere [5,12,42,43]. Based on the previous preclinical and clinical trial results and general consensus from investigational *S. aureus* vaccine candidates, we designed and developed a rFSAV adopting the following strategies:

First, many studies have shown that the pathogenesis of *S. aureus* is so complex that a single antigen cannot provide sufficient protection [6,43]. We designed a “cocktail” vaccine formulation for multiple targets. These antigens including bacterial toxins, immune escape factor and proteins closely associated with bacterial growth and metabolism, played key roles in pathogenesis of *S. aureus*. In addition, we mutated and modified these antigens to remove harmful toxic effects and maintain good immunogenicity. SpA was mutated to inhibit the non-specific binding of SpA to the Fc fragment of IgG. SEB was mutated to remove the enterotoxin activity of SEB and Hla was mutated to remove its haemolytic activity. Furthermore, IsdB was truncated and fused with mHla to clear away the potential toxicity and retained their strong immunogenicity.

Second, although the animal experiment has limitations in evaluating the efficacy of the vaccine for *S. aureus* [44], the animal model, based on characteristics of clinical *S. aureus* infection, is still an indispensable evaluation method for preclinical study of the vaccine. Therefore, the mouse sepsis model and pneumonia model were established mimicked the infection pathways and pathogenic mechanisms of *S. aureus*, which were subsequently applied to the evaluation of the immunogenicity, immune-protection and cross-protection of rFSAV. This provided an experimental important reference for the selection of target populations in the follow-up clinical trials.

Third, we designed an optimal immunization procedure (days 0, 3 and 7) for rFSAV to satisfy clinical needs of high-risk target groups in hospital referring to the immunization procedure of rabies vaccine. Our results indicated that the antibody titre produced by this immunization procedure was increased by fourfold on D7 after primary immunization. The survival results in the sepsis mouse model showed that rFSAV induced a good immune protection on D8 after primary immunization, that was no significant difference compared with D18 and D68. This indicates that the unconventional “perioperative” immunization procedure may be more suitable for clinical practice.

Most *S. aureus* vaccine candidates already in clinical trials are designed to elicit robust antibody responses, but it turns out that just considering functional antibodies against *S. aureus* is insufficient [18,19]. Of course, it is undeniable that a robust level of vaccine-induced antibodies are important [18,19]. In our study, we also focused on the role of rFSAV-induced opsonic and neutralizing antibodies (Figure 5 and Figure 6). However, protective immunity from *S. aureus* infection is not defined and so it may be a better strategy that a *S. aureus* vaccine can induce a comprehensive immune response. rFSAV includes an aluminum phosphate adjuvant, which can be recognized by the innate immune system leading to multiple downstream effects [45,46], such as activation of the inflammasome and the type 2 innate response [45], stimulation and differentiation of CD4^+^ T cells and so on [46]. In our research, we also found that the aluminum adjuvant provided a slight protective effect against *S. aureus* infection (Figure 2C–G and Figure 3A,K–N). What is more, we also detected that rFSAV can induce Th1 and Th17 cell immune responses (Figure 4A). Although these detailed mechanisms are very complicated and undefined, it is necessary that inducing comprehensive natural and acquired cellular and humoral immune responses for an effective *S. aureus* vaccine.

V710 was developed by Merck as a genetically engineered recombinant subunit vaccine and its main component is IsdB full length (including N1 and N2 domains) expressed as inclusion body form in yeast [8,9]. Their results of clinical trials do not support the use of the V710 vaccine for patients undergoing surgical interventions [8]. A retrospective study of V710’s Phase IIb/III trial showed that the coincidence of 3 factors (low prevaccination IL2 levels, receipt of V710 and postoperative *S. aureus* infection) appeared to substantially increase mortality in their study population after major cardiothoracic surgery [47]. In the animal safety evaluation test of rFSAV, the 40 cynomolgus monkeys were immunized with rFSAV at 3 times dose, and no toxicological changes were observed both in the rFSAV group or the placebo group. In particular, there were no significant differences in lymphocyte subset distribution and serous cytokine levels (IL-4, IL-5, TNF-α, IFN-γ, IL-2 and IL-6) between the vaccine and placebo groups (data unpublished). Although the pathophysiological analysis results to V710 and host factors were still hypothesis-generating and speculative, we will continue to concern that the impact of immune predispositions on the safety and efficacy of rFSAV in the following clinical trials.

## 5. Conclusions

rFSAV is a potentially promising vaccine candidate for defensing against *S. aureus* infection.

## Figures and Tables

**Figure 1 vaccines-08-00134-f001:**
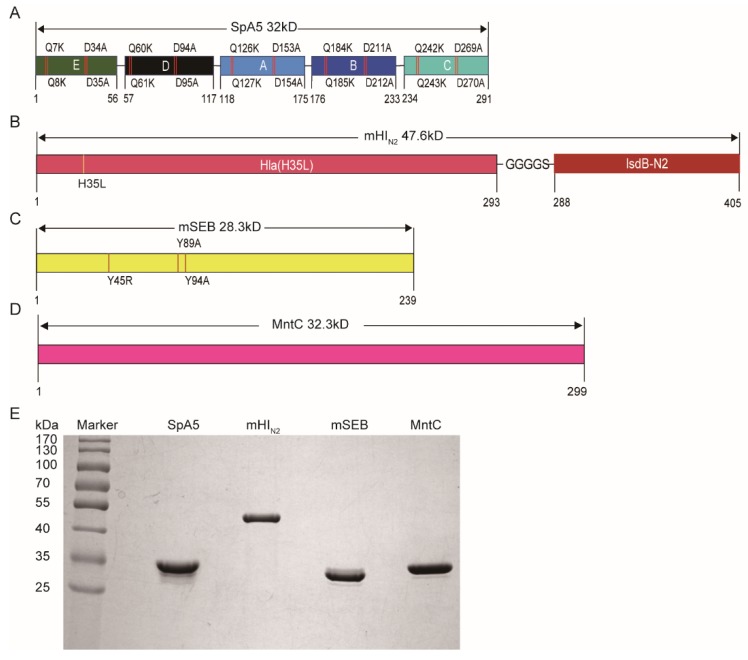
Antigen selection, modification and preparation. (**A**–**D**) Schematic diagram illustrating the primary structure of staphylococcal protein A (SpA)5, mHI_N2_, mSEB and manganese transport protein C (MntC), respectively. (**E**) SpA5, mHI_N2_, mSEB and MntC were purified and analyzed by SDS-PAGE.

**Figure 2 vaccines-08-00134-f002:**
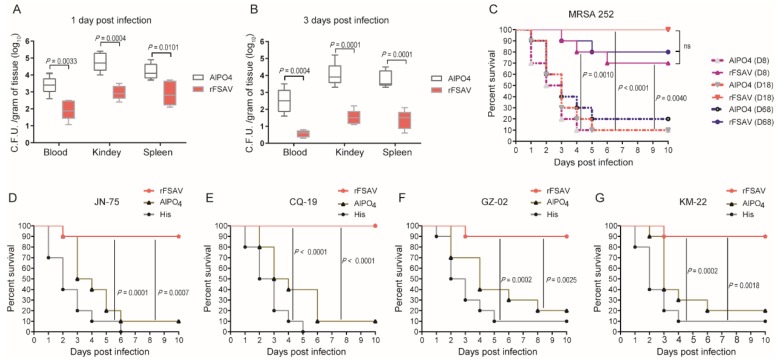
rFSAV stimulates protective immunity in a *S. aureus* sepsis model. (**A**,**B**) Efficacy of rFSAV on the spread of *S. aureus*. BALB/c mice (*n* = 10) were immunized with rFSAV and challenged with MRSA252 at 3.0 × 10^8^ CFUs/mouse by tail intravenous injection. The number of viable bacteria in the blood, kidney and spleen of mice (*n* = 10) at 1 and 3 days post-infection were shown. Data were presented in box and whisker plots, and the medians are shown. Differences were compared to determine their significance using Student’s *t*-test. (**C**) BALB/c mice (*n* = 10) were immunized with rFSAV and challenged with MRSA at 6.0 × 10^8^ CFUs/mouse on 8, 18 and 68 days post the first immunization by tail intravenous injection. (**D**–**G**) BALB/c mice (*n* = 10) were immunized with rFSAV and challenged with JN-75, CQ-19, GZ-02 and KM-22 (3.0 × 10^8^, 5.0 × 10^8^, 7.0 × 10^8^ and 2.0 × 10^8^ CFUs/mouse, respectively) on 18 days. (**C**–**G**) The survival rate was monitored for 10 days. The *p*-values were calculated using the Mantel-Cox log-rank test.

**Figure 3 vaccines-08-00134-f003:**
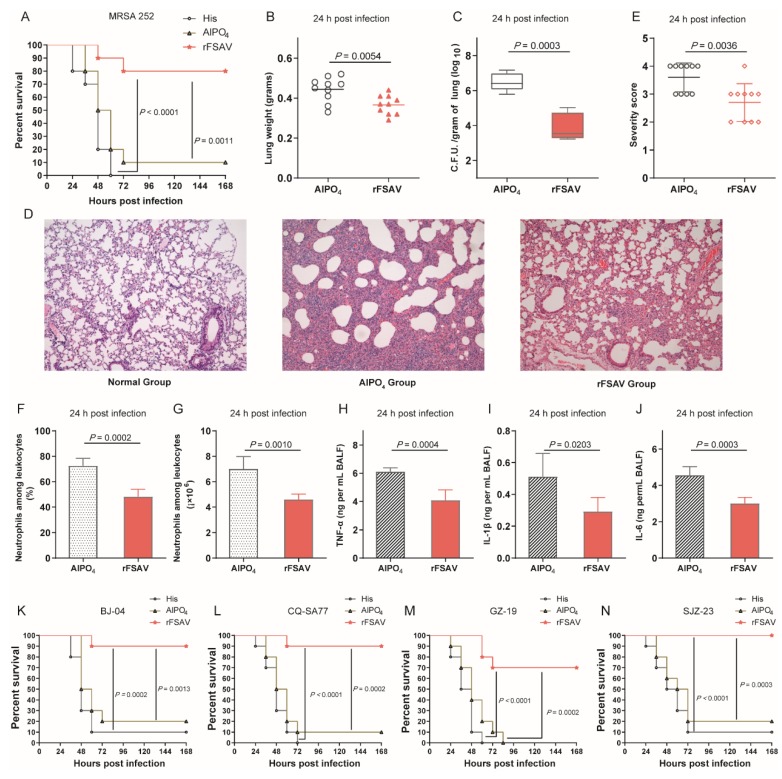
Protective efficacy of rFSAV in a murine *S. aureus* pneumonia model. (**A**) Immunized and control C57BL/6 mice (*n* = 10) were challenged with MRSA252 at 1.0 × 10^8^ CFUs/mouse by intratracheal injection. (B–J) The immunized mice and control mice were infected intratracheally with MRSA252 at 5.0 × 10^7^ CFUs/mouse. (**B**) Lungs in infected mice (*n* = 10) immunized rFSAV were weighed. The data were presented as scatter plots. (**C**) The number of viable bacteria in the lungs of mice (*n* = 10) at 24 h post infection was shown. Data were presented in box and whisker plots, and the medians were shown. (**D**) Hematoxylin-eosin staining of lungs from mice 24 h post infection. Representative histopathological sections from 10 mice per group were shown (magnification = 100×). (**E**) Semi-quantification of lung inflammation. Severity scores of lungs (*n* = 10) from mice 24 h post infection were shown. The data were presented as scatter plots. (**F**,**G**) Evaluation of neutrophil infiltration in infected mice (*n* = 10). The bar represented the percentage (**F**) and the number (**G**) of neutrophils in the BALF of mice at 24 h post challenge. (**H**–**J**) Quantitative detection of proinflammatory cytokines TNF-α, IL-1β and IL-6 in infected mice (*n* = 10). The data (**F–J**) were shown as the means ± SD. (**K**–**N**) Immunized and control C57BL/6 mice (*n* = 10) were challenged with BJ-04, CQ-SA77, GZ-19 and SJZ-23 (9.0 × 10^8^, 4.0 × 10^8^, 3.0 × 10^8^ and 4.0 × 10^8^ CFUs/mouse, respectively) on 18 days. (**A**,**K**–**N**) The survival rates were recorded every 12 h over a 7-day observation period post challenge. The *p*-value was calculated using the Mantel-Cox log-rank test. The differences (**B**,**C**,**E**–**J**) were compared using Student’s *t*-test.

**Figure 4 vaccines-08-00134-f004:**
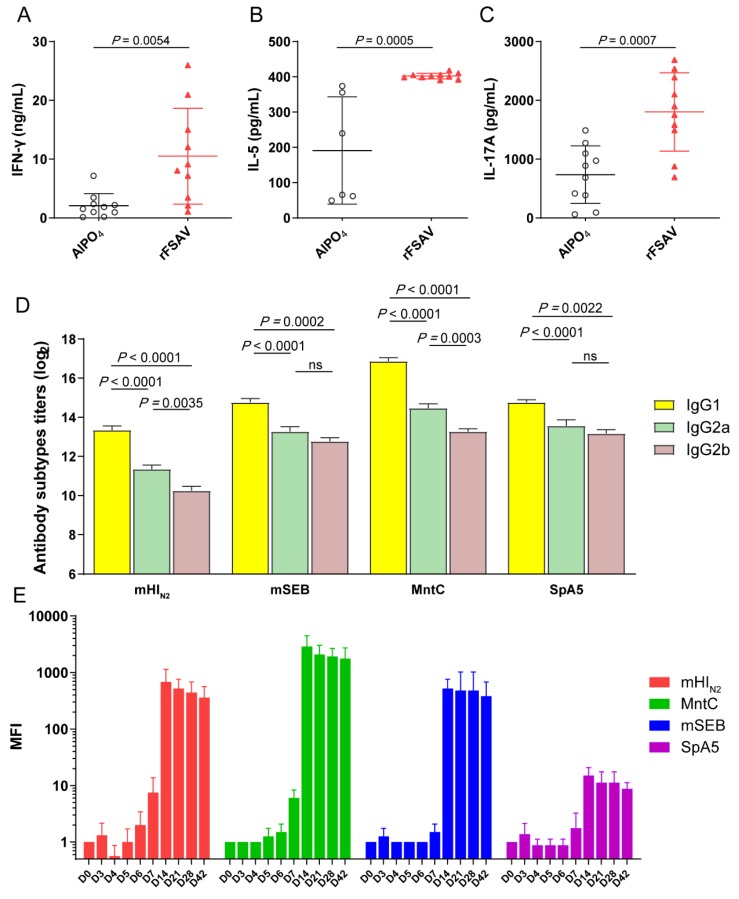
Analysis of cytokine and antibody responses. (**A**–**C**) Comparison of cytokine production by antigens stimulated splenocytes from immunized and control mice. 10 days after the final immunization, spleens (*n* = 5) were processed and stimulated recombinant proteins, and the levels of IFN-γ, IL-5 and IL-17A in each culture supernatant were measured after 72 h. The differences were compared using Student’s *t*-test. (**D**) Comparison of serum IgG1, IgG2a and IgG2b subtypes among antigens in the immunized mice (*n* = 5). Serum was obtained at 10 days after the final immunization, and the levels of IgG1, IgG2a and IgG2b subtypes were expressed as the mean of log_2_ titers. (**E**) The specific IgG of mHI_N2_, MntC, mSEB and SpA5 in mice immunized with rFSAV was detected at each time point from days 0 to 42, and the levels of each antigen specific IgG were expressed as Mean Fold Increase (MFI). (**D**,**E**) Data were shown as the mean ± SEM. The *p*-values were calculated using one-way ANOVA (ns = no significance).

**Figure 5 vaccines-08-00134-f005:**
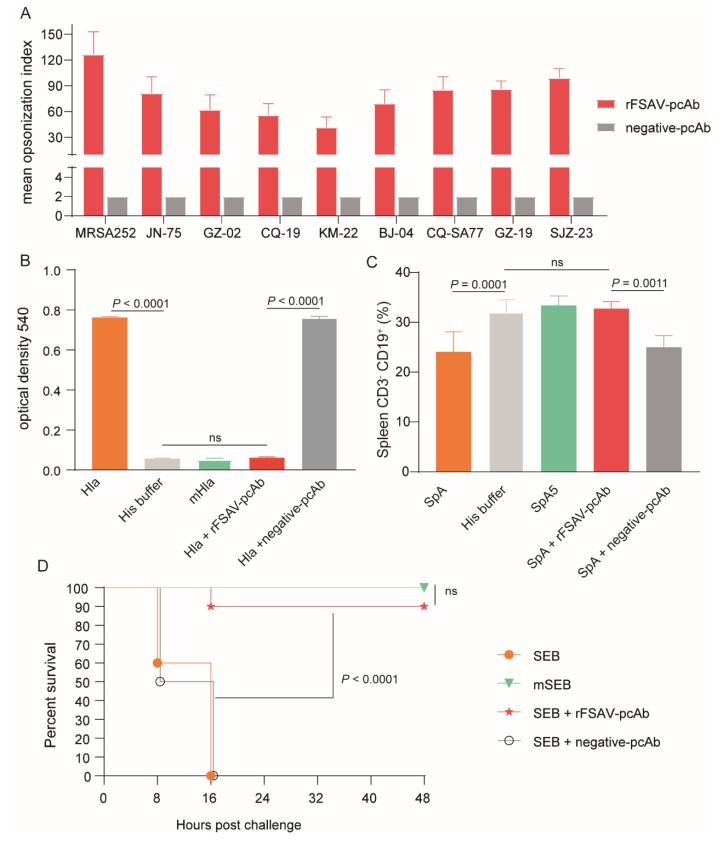
Analysis of rFSAV-pcAb’s efficacy on promoting opsonophagocytosis and neutralizing toxins. (**A**) Comparative analysis of opsonophagocytic killing activity against different clinical isolations of *S. aureus* by rFSAV-pcAb. The mean opsonization index was calculated to determine killing activity. The data were shown as the mean ± SD derived from three independent experiments. (**B**–**D**) Hla, SpA or SEB were incubated with rFSAV-pcAb or negative-pcAb at 37 ℃ for 30 min prior to the start of the study, respectively. (**B**) Hemolytic activity assay. 1% Rabbit erythrocytes were incubated with Hla pre-incubated with rFSAV-pcAb or negative-pcAb at 37 ℃ for 30 min. The supernatant containing hemoglobin was measured spectrophotometrically at 540 nm to detect the hemolytic activity. The data were shown as the mean ± SD derived from three independent experiments. The *p*-values were calculated using one-way ANOVA (ns = no significance). (**C**) B cell superantigen activity assay. CD3^−^ CD19^+^ B cells in splenic tissue of BALB/c mice (*n* = 5) treated with SpA pre-incubated with rFSAV-pcAb or negative-pcAb were quantified by FACS. The data were shown as the mean ± SD, and the *p*-values were calculated using one-way ANOVA. (**D**) Intestinal toxin activity assay. mice were injected i.p. with D-Galactosamine (200 mg/mL) followed by an intramuscular injection with SEB pre-incubated with rFSAV-pcAb or negative-pcAb. The survival rates were recorded every 8 h over a 2-day observation period post challenge. The Mantel-Cox log-rank test was used to calculate *p-*value.

**Figure 6 vaccines-08-00134-f006:**
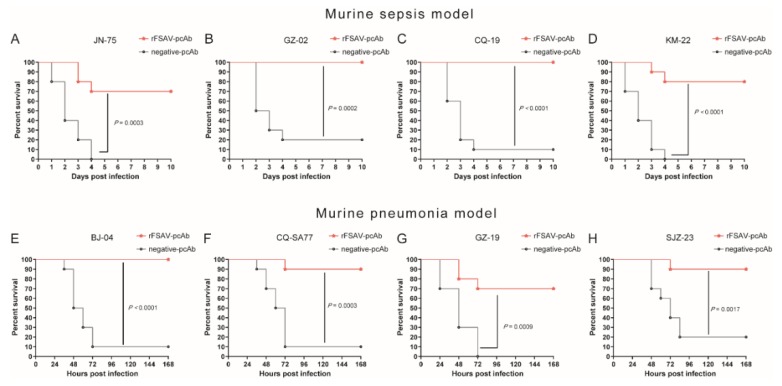
rFSAV-pcAb broadly protected mice from different clinical *S. aureus* strains challenge in murine sepsis and pneumonia models. (**A**–**D**) Protection of passive immunization with rFSAV-pcAbs in the mouse sepsis model. BALB/c mice (*n* = 10) were injected intravenously with 100 μL of rFSAV-pcAbs or negative-pcAbs (20 mg/mL). Two hours later, the mice were challenged with JN-75, CQ-19, GZ-02 or KM-22 (3.0 × 10^8^, 5.0 × 10^8^, 7.0 × 10^8^ or 2.0 × 10^8^ CFUs/mouse, respectively) by tail intravenous injection. The survival rate was monitored for 10 days. (**E**–**H**) Protection of passive immunization with rFSAV-pcAbs in the mouse pneumonia model. C57BL/6 mice (*n* = 10) were injected intravenously with 100 μL of rFSAV-pcAbs or negative-pcAbs (20 mg/mL). Two hours later, the mice were challenged with BJ-04, CQ-SA77, GZ-19 or SJZ-23 (9.0 × 10^8^, 4.0 × 10^8^, 3.0 × 10^8^ or 4.0 × 10^8^ CFUs/mouse, respectively) by intratracheal injection. The survival rates were recorded every 12 h over a 7-day observation period post challenge. (**A**–**H**) The Mantel-Cox log-rank test was used to compare differences between passive immunization and control groups.

**Table 1 vaccines-08-00134-t001:** The survival rates of recombinant five-antigen *S. aureus* vaccine (rFSAV) and control groups through six repetitions in a *S. aureus* sepsis model.

Groups		Survival Rate (10 days)						
		V1	V2	V3	V4	V5	V6	Mean
Control	His buffer	0	10%	10%	20%	10%	30%	16%
	AlPO_4_	30%	25%	20%	30%	20%	30%	25%
Monovalent	mHI_N2_	10%	30%	40%	30%	25%	30%	27%
	mSEB	30%	11%	30%	30%	30%	38%	24%
	SpA5	70%	20%	30%	35%	40%	35%	40%
	MntC	56%	0%	30%	30%	35%	27%	29%
Bivalent	mHI_N2_ + mSEB	45%	38%	30%	60%	80%	90%	60%
	mHI_N2_ + SpA5	58%	40%	50%	60%	80%	80%	62%
	mHI_N2_ + MntC	60%	50%	25%	80%	90%	100%	69%
	mSEB + SpA5	76%	90%	60%	80%	100%	60%	78%
	mSEB + MntC	65%	70%	70%	78%	60%	89%	73%
	SpA5 + MntC	73%	70%	80%	80%	80%	90%	80%
Trivalent	mHI_N2_ + mSEB + SpA5	76%	100%	78%	67%	67%	80%	78%
	mHI_N2_ + mSEB + MntC	71%	60%	50%	80%	60%	100%	70%
	mHI_N2_ + SpA5 + MntC	70%	80%	89%	67%	70%	60%	73%
	mSEB + SpA5 + MntC	80%	70%	90%	80%	89%	90%	84%
rFSAV	mHI_N2_ + mSEB + SpA5 + MntC	90%	80%	90%	85%	85%	90%	87%
V710 [24]	IsdB	70%	55%	50%	-	-	-	58%

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
