# Peer review of "Rapid and Broad Immune Efficacy of a Recombinant Five-Antigen Vaccine against Staphylococcus aureus Infection in Animal Models"

_vaccines, 2020, doi:10.3390/vaccines8010134_

Round 1
Reviewer 1 Report
Zeng et al. present the development of a recombinant five-antigen S. aureus vaccine (rFSAV) which showed broad protection against several strains of S. aureus including MRSA252. The authors validate the efficacy of this vaccine in an S. aureus lethal sepsis model and pneumonia mouse model, and also propose the underlying mechanism of vaccine action. In addition, the study reports a better survival rate in S. aureus challenged mice that were vaccinated with rFSAV compared with the Merck V710 vaccine, which has undergone clinical trial and efficacy studies. The findings reported here, therefore, are of significance considering the need for effective vaccines that can counter severe S. aureus infections, including drug-resistant strains.
Below are some specific concerns that I feel the authors must address.
Specific comments:
- Figure 1: Antigen purity is mentioned as 95% based on SDS-PAGE. Was a silver stain done to determine this? Could the authors include the HPLC data as well? The purity of the preparation is important, hence care must be shown to prove that point.
- Table 1: V710 was tested in only 3 repetitions for survival rate in Table 1, whereas all other treatments were conducted in 6 different experiments. Why? The variations within each of the 6 repetitions for the combinations tested are broad enough to warrant the testing of V710 in 6 different experiments as well if any conclusions as to the efficacy of rFSAV over V710 is to be drawn.
- Figure 4: The figure listing in the legends is not compatible with the figures.
- Lines 385-395: The authors need to describe this section in a better way. The results from different assays are described in a succinct manner, but it would be better if a brief assay description is provided before to orient the readers to the conclusions drawn. I acknowledge that the legends of some figures describe these assays in some detail, but having that description within the results text would be nice. Also, the conclusions from each treatment within experiments, e.g. SPA5 in Fig. 5C and mSEB in 5D should also be reported to provide a more complete analysis of the results.
- Figure 5A, and 6A-B: Why wasn't MRSA252 tested? Efficacy of the rFSAV against antibiotic-resistant strains is an important point in this paper.
Minor points:
- Line 227: "haeme"
- Line 369-70: It's probably not required to mention that the data is a scatter plot.
- Line 385: ....rFSAV-pcAb aim to S, aureus...?
- Line 392: ....all mice that "were" challenged with....
- Line 444: Grammatical correction is required.
- Line 451: ". And Hla.."?
- Line 453: "...and retained the strongly immunogenecity."?
Author Response
Response to reviewers’ comments
Paper #vaccines-732821
Dear reviewers,
We authors appreciate you very much for your valuable and constructive comments on our manuscript. These comments are necessary for us to improve the overall quality of our manuscript. We have revised our manuscript accordingly, the response are listed as follows:
Response to Comments of Reviewer #1:
Specific comments:
- Figure 1: Antigen purity is mentioned as 95% based on SDS-PAGE. Was a silver stain done to determine this? Could the authors include the HPLC data as well? The purity of the preparation is important, hence care must be shown to prove that point.
Thanks for your suggestion. Both silver stain and Coomassie blue stain in SDS-PAGE can be used to determine purity of protein referring to the ICH Q4B related documents. The purity of all 4 proteins (SpA5, mHIN2, mSEB and MntC) has been determined by SDS-PAGE (including Coomassie blue stain and silver stain) and high performance liquid chromatography respectively during the quality research of rFSAV. The results of all these methods showed that the purity of all 4 proteins was more than 95%, which met the standard of China Food and Drug Administration. We only show the results of Coomassie blue staining, because the separation principle of Coomassie blue stain and silver stain is the same. The results of silver stain were shown as follow. Two bands appeared in the figure of mSEB due to the isomerization of disulfide bonds, which was analyzed by Liquid chromatography- mass spectrometry (Electrospray ionization), disulfide bond analysis and HPLC analysis.
According to your opinion, the HPLC data was included in supplementary materials. To see Supplementary Figure 2 and line 250.
- Table 1: V710 was tested in only 3 repetitions for survival rate in Table 1, whereas all other treatments were conducted in 6 different experiments. Why? The variations within each of the 6 repetitions for the combinations tested are broad enough to warrant the testing of V710 in 6 different experiments as well if any conclusions as to the efficacy of rFSAV over V710 is to be drawn.
We are so sorry that we didn't describe it clearly. We cannot obtain the same quality V710 vaccine as Merck, because V710 is not available on the market. The data about survival rates of V710 vaccination is from Figure 1 in the literature as follow.
Kuklin, N.A.; Clark, D.J.; Secore, S.; Cook, J.; Cope, L.D.; McNeely, T.; Noble, L.; Brown, M.J.; Zorman, J.K.; Wang, X.M., et al. A novel Staphylococcus aureus vaccine: iron surface determinant B induces rapid antibody responses in rhesus macaques and specific increased survival in a murine S. aureus
We have explained this in the results section. To see lines 264-265.
- Figure 4: The figure listing in the legends is not compatible with the figures.
That is our fault. We have checked and corrected the legends of Figure 4 thoroughly. To see lines 372-383.
- Lines 385-395: The authors need to describe this section in a better way. The results from different assays are described in a succinct manner, but it would be better if a brief assay description is provided before to orient the readers to the conclusions drawn. I acknowledge that the legends of some figures describe these assays in some detail, but having that description within the results text would be nice. Also, the conclusions from each treatment within experiments, e.g. SPA5 in Fig. 5C and mSEB in 5D should also be reported to provide a more complete analysis of the results.
Thanks for your suggestion. According to your opinion, we have revised this section. We have added a more complete analysis of these results to draw the conclusions. To see lines 402-422.
- Figure 5A, and 6A-B: Why wasn't MRSA252 tested? Efficacy of the rFSAV against antibiotic-resistant strains is an important point in this paper.
Thanks for your suggestion. MRSA252 was a reference strain in this study, and the sequences encoding SpA5, mHIN2 and MntC were all from the genome of S. aureus MRSA252. Also, MRSA252 had been repeatedly evaluated during the exploration and establishment of the opsonophagocytic killing assay (Figure 5A) and passive immunization of rFSAV-pcAb test (Figure 6A-6B). Of course it also showed very good results. Because the results mainly exhibited the efficacy of the rFSAV-specific antibodies against clinical S. aureus strains which included antibiotic-resistant strains (Supplementary Table 1) in vivo and in vitro, we did not show the results about rFSAV-pcAb against MRSA252. According to your opinion, we have added the result of opsonophagocytic killing assay against MRSA252 in Figure 5A, and the result of passive immunization protects mice against MRSA252 in murine sepsis and pneumonia models in Supplementary Figure S3. To see Figure 5A and Supplementary Figure S2, and lines 387-389, 443-446.
Minor points:
- Line 227: "haeme"
Thanks for your suggestion. But according to the references (35. Unique heme-iron coordination by the hemoglobin receptor IsdB of Staphylococcus aureus. Biochemistry 2011, 50, 5443-5452, doi:10.1021/bi200369p.), we think “heme” is more suitable. To see line 233.
- Line 369-70: It's probably not required to mention that the data is a scatter plot.
Thanks for your suggestion. According to your opinion, we have revised this section and made it more concise. To see lines 375-376.
- Line 385: ....rFSAV-pcAb aim to S, aureus...?
We have revised this sentence ( “on” instead of “aim to” ). To see line 392.
- Line 392: ....all mice that "were" challenged with....
We have revised this sentence. To see line 418.
10 Line 444: Grammatical correction is required.
Thanks for your suggestion. We have revised this sentence. To see line 475.
- Line 451: ". And Hla.."?
Thanks for your suggestion. We have revised this sentence. "and" was with small letter and the "." was removed. To see line 482.
- Line 453: "...and retained the strongly immunogenecity."?
We have revised this sentence. “their strong” was instead of “the strongly”. To see line 484.
Reviewer 2 Report
The manuscript entitled "Rapid and broad immune efficacy of a recombinant five-antigen vaccine against S. aureus infection in animal model" used a cocktail of highly conserved secreted/surface anchored proteins (e.g. toxins/virulence) to develop a recombinant vaccine against SA.
Introduction:
- There are several typos/grammer mistake. e.g. Line 48: "and" should be with small letter and the "." should be removed. There are several others similar mistakes in the text as well.
- There are several studies using S. aureus membrane vesicles that provide against SA. These studies should be cited as they are kind of mix of several antigens including the antigens selected by the authors.
- In the last paragraph (line 63 and 64) it has been stated that a large number of screening test has been performed to select the antigens. However, in the results sections it has been stated that these antigens are selected based on a literature review. This sentence MUST be corrected.
Materials and Methods:
The immunisation strategy should be described in more details. In addition showing a diagram represents timeline for vaccination procedure and SA challenge in the results section/supplementary section would be very helpful.
The section 2.13 must be briefly explained.
Results:
- The graph for IgG subclasses response should be updated and present at multiple doses of serum dilution
- Opsonophagocytosis performed with cell lines instead of primary cells (purified PMNs from whole blood). HL-60 cells are not always properly differentiated and no markers has been checked. This experiment must be repeated.
Discussion:
The references are not obtained and the results are not properly discussed.
Author Response
Response to reviewers’ comments
Paper #vaccines-732821
Dear reviewers,
We authors appreciate you very much for your valuable and constructive comments on our manuscript. These comments are necessary for us to improve the overall quality of our manuscript. We have revised our manuscript accordingly, the response are listed as follows:
Response to Comments of Reviewer #2:
Introduction:
- There are several typos/grammer mistake. e.g. Line 48: "and" should be with small letter and the "." should be removed. There are several others similar mistakes in the text as well.
Thanks for your suggestion. We have checked and corrected the manuscript thoroughly. To see lines 48, 52, 262, 376, 382, 482.
- There are several studies using S. aureus membrane vesicles that provide against SA. These studies should be cited as they are kind of mix of several antigens including the antigens selected by the authors.
Thanks for your suggestion. Several studies have shown that membrane-derived vesicles (MVs) of S. aureus can provide protective immunity in murine models of infection. However, due to difficulties in quality control and other reasons, no clinical trials have been conducted for MVs so far. In the introduction section of our study, we mainly introduce S. aureus vaccines that have already undergone clinical trials, and our main research area is recombinant proteins of S. aureus. For these reasons, the literature of MVs of S. aureus is not cited.
- In the last paragraph (line 63 and 64) it has been stated that a large number of screening test has been performed to select the antigens. However, in the results sections it has been stated that these antigens are selected based on a literature review. This sentence MUST be corrected.
Although these antigens are selected based on a literature review, but the protective antigen combinations were selected by large number of screening test. In fact, in our previous research, we selected dozens of antigens from S. aureus and screened hundreds of combinations to obtain such a highly protected "cocktail" vaccine. Of course, if you insist on correcting this sentences, we will follow your suggestions.
Materials and Methods:
- The immunisation strategy should be described in more details. In addition showing a diagram represents timeline for vaccination procedure and SA challenge in the results section/supplementary section would be very helpful.
Thanks for your suggestion. According to your opinion, we have added a diagram that represented timeline for vaccination procedure and S. aureus in the supplementary section. To see Supplementary Figure S1.
- The section 2.13 must be briefly explained.
Thanks for your suggestion. According to your opinion, we have briefly explained Hemolytic activity assay and intestinal toxin activity assay in the section 2.13. To see lines 188-194.
Results:
- The graph for IgG subclasses response should be updated and present at multiple doses of serum dilution.
Thanks for your suggestion. However, both methods that the graph for IgG subclasses response is present at multiple doses of serum dilution or titers are acceptable. The method we used is based on the following reliable references:
(1) Zuo, Q.F.; Yang, L.Y.; Feng, Q.; Lu, D.S.; Dong, Y.D.; Cai, C.Z.; Wu, Y.; Guo, Y.; Gu, J.; Zeng, H., et al. Evaluation of the protective immunity of a novel subunit fusion vaccine in a murine model of systemic MRSA infection. PLoS One 2013, 8, e81212, doi:10.1371/journal.pone.0081212.
(2) Chen, Y.; Yang, F.; Yang, J.; Hou, Y.; He, L.; Hu, H.; Lv, F. Aluminum (oxy) Hydroxide Nanorods Activate an Early Immune Response in Pseudomonas aeruginosa Vaccine. ACS Appl Mater Interfaces 2018, 10, 43533-43542, doi:10.1021/acsami.8b18164.
(3) Yang, F.; Gu, J.; Yang, L.; Gao, C.; Jing, H.; Wang, Y.; Zeng, H.; Zou, Q.; Lv, F.; Zhang, J. Protective Efficacy of the Trivalent Pseudomonas aeruginosa Vaccine Candidate PcrV-OprI-Hcp1 in Murine Pneumonia and Burn Models. Sci. Rep. 2017, 7, 3957, doi:10.1038/s41598-017-04029-5.
What’ more, the data of IgG subclasses response is expressed in the form of titers, which is more convenient for statistical analysis.
- Opsonophagocytosis performed with cell lines instead of primary cells (purified PMNs from whole blood). HL-60 cells are not always properly differentiated and no markers has been checked. This experiment must be repeated.
Thanks for your suggestion. HL-60 cells instead of primary cells used in opsonophagocytic killing assay is based on the following reasons:
(1) The method we used is based on the following reliable references:
- Begier, E.; Seiden, D.J.; Patton, M.; Zito, E.; Severs, J.; Cooper, D.; Eiden, J.; Gruber, W.C.; Jansen, K.U.; Anderson, A.S., et al. SA4Ag, a 4-antigen Staphylococcus aureus vaccine, rapidly induces high levels of bacteria-killing antibodies. Vaccine 2017, 35, 1132-1139, doi:10.1016/j.vaccine.2017.01.024.
- Pozzi, C.; Wilk, K.; Lee, J.C.; Gening, M.; Nifantiev, N.; Pier, G.B. Opsonic and protective properties of antibodies raised to conjugate vaccines targeting six Staphylococcus aureus antigens. PloS one 2012, 7, e46648, doi:10.1371/journal.pone.0046648.
- Nanra, J.S.; Buitrago, S.M.; Crawford, S.; Ng, J.; Fink, P.S.; Hawkins, J.; Scully, I.L.; McNeil, L.K.; Aste-Amezaga, J.M.; Cooper, D., et al. Capsular polysaccharides are an important immune evasion mechanism for Staphylococcus aureus. Human vaccines & immunotherapeutics 2013, 9, 480-487, doi:10.4161/hv.23223.
(2) The differentiation and detection of HL-60 cells were according to the protocol for UAB-MOPA (https://www.vaccine.uab.edu/uploads/mdocs/UAB-MOPA.pdf), which was recommended by WHO. In this protocol, differentiated HL60 cells will be considered acceptable as effector cells if all of the following criteria are met:
- Viability is ≥90% (trypan blue) or ≥65% (propidium iodide).
- CD35 is expressed on ≥55% of the cells.
- CD71 is expressed on ≤20% of the cells.
- Apoptotic cells, defined as Annexin V+/PI-, represent ≤25% of the cells.
What’ more, we determined the HL-60 cells phenotype of both differentiated cells and undifferentiated cells in the opsonophagocytic killing assay. The results are shown as follow.
HL-60 cells were stained using the following antibodies:
- PE anti-human CD35 Antibody(Biolegend,Catalog Number 333405);
- PE Mouse IgG1, κ Isotype Ctrl (FC) Antibody(Biolegend,Catalog Number 400113);
- APC anti-human CD71 Antibody(Biolegend,Catalog Number 334107);
- APC Mouse IgG2a, κ Isotype Ctrl Antibody(Biolegend,Catalog Number 400219);
- Propidium Iodide Staining Solution(BD Pharmingen,Catalog Number 556463);
- FITC Annexin V(BD Pharmingen,Catalog Number 556420)。
Obviously, The HL-60 cells we used meet the criteria, so the differentiated HL60 cells are acceptable as effector cells in opsonophagocytic killing assay.
Discussion:
- The references are not obtained and the results are not properly discussed.
Thanks for your suggestions. We have revised the section of discussion thoroughly. Especially to prove our conclusion based on our data. To see lines 499-513.
Round 2
Reviewer 2 Report
Some of the concerns has been addressed. However, the screening test and selection process is an important part of the novelty of this study. (1) That sentence in the results section should be edited (2) if the screening strategy and selection have been published by authors in the past, that reference should be cited. If they have not published the selection/screen strategy yet, a new section in the mat-met should describe the selection and screening strategy.
Author Response
Response to reviewers’ comments Paper #vaccines-732821
Dear reviewers,
We authors appreciate you very much for your valuable and constructive comments on our manuscript. These comments are necessary for us to improve the overall quality of our manuscript. We have revised our manuscript accordingly, the response are listed as follows:
Response to Comments of Reviewer #2:
Some of the concerns has been addressed. However, the screening test and selection process is an important part of the novelty of this study. (1) That sentence in the results section should be edited (2) if the screening strategy and selection have been published by authors in the past, that reference should be cited. If they have not published the selection/screen strategy yet, a new section in the mat-met should describe the selection and screening strategy.
Thanks for your suggestion. We did obtain the rFSAV for multiple targets through a large number of screening tests of protective antigen combinations. The selection and screening strategy were as follow:
First, a candidate antigen library of Staphylococcus aureus was established through literature review (including preclinical and clinical studies) and bioinformatics analysis.
Second, the candidate antigens in the library were designed at the molecular level, and the methods mainly included fragments truncated, selection of active centers, non-toxic mutations, fusion expression and so on.
Third, these designed antigens were cloned and expressed in the prokaryotic expression system, and then they were purified one by one.
Fourth, Different combinations of these antigens were designed, which contained
bivalent combinations, trivalent combinations, and quadrivalent combinations. Further, the immunogenicity and immune efficacy of these designed combinations were evaluated in animal experiments.
Finally, these antigen combinations were given a score according to established criteria based on the stability of antigen preparation process, the quality of the antigens, the immunogenicity of antigen combinations and the protection rates of antigen combinations. So we selected several promising antigen combinations, of which rFSAV was one.
Therefore, there are a lot of unpublished results, which included: (1) Molecular design processes of antigens, such as primer sequences, mutation site and so on. (2) The expression and purification figures of antigens. (3) Quality inspection results of antigens, such as purity, molecular weight, endotoxin assay and so on. (4) Geometric mean titers of each antigen in various antigen combinations. (5) Survival curves and protection rates of different antigen combinations. The table below shows all the single and bivalent fusion antigens in the antigen library.
Because this article focuses on evaluating rFSAV, which contains five antigen targets, including SpA, Hla, IsdB-N2, SEB and MntC, the previous selection and screening strategy is not discussed in detail in this article. According to your suggestion, we have deleted the description of screening process of the antigen combinations in introduction in this study. To see lines 62-64.
Number |
Gene name |
Protein name |
nucleotide (bp) |
amino acid (aa) |
1 |
ClfA |
Clumping factor A |
3090 |
1029 |
2 |
Clfa1 |
33-213AA of clfa |
543 |
181 |
3 |
Clfa2 |
40-309aa of clfa |
780 |
260 |
4 |
CNA1 |
collagen adhesin |
1209 |
403 |
5 |
CNA2 |
collagen adhesin |
1770 |
590 |
6 |
Cog |
staphylocoagulase |
1410 |
470 |
7 |
FnbA |
fibronectin-binding protein A |
2898 |
965 |
8 |
FnbA1 |
Fragment of FnbA |
1419 |
473 |
9 |
FnbA2 |
Fragment of FnbA |
1329 |
443 |
10 |
FnbA2-8 |
Fragment of FnbA |
636 |
212 |
11 |
FnbA2-8M |
Fragment of FnbA(mutant) |
636 |
212 |
12 |
GrfA |
ATP-binding protein |
1602 |
533 |
13 |
Hla |
α-toxin |
960 |
319 |
14 |
mHla(H35L) |
non-toxic mutant |
960 |
319 |
15 |
mHla(delta110-150) |
non-toxic mutant |
837 |
279 |
16 |
IsdB |
Iron surface determinant B |
1791 |
596 |
17 |
IsdB1 |
I1 domain of IsdB |
381 |
127 |
18 |
IsdB2 |
I2 domain of IsdB |
357 |
119 |
19 |
MntC |
Manganese Transport Protein C |
855 |
285 |
20 |
RAP |
rRNA-binding protein |
831 |
277 |
21 |
TRAP |
target of RNAIII-activating protein |
501 |
167 |
22 |
SAR1404 |
ATP-binding protein |
1581 |
527 |
23 |
SEB |
staphylococcal enterotoxin B |
786 |
261 |
24 |
mSEB (L45R, Y89A, Y94A) |
non-toxic mutant |
786 |
261 |
25 |
SEC |
staphylococcal enterotoxin C |
756 |
251 |
26 |
SpA |
Staphylococcus aureus Protein A |
1242 |
414 |
27 |
vWBP |
von Willebrand actor-binding protein |
1443 |
481 |
28 |
SPAD(KA) |
mutant of D domain of SPA |
150 |
50 |
29 |
SpA5(EDABC) |
5 domains of SPA |
873 |
291 |
30 |
Clfa2-mSEB(RAA) |
Bivalent fusion protein |
1581 |
527 |
31 |
F2-8M- SpA55 |
Bivalent fusion protein |
1524 |
508 |
32 |
SEB(RAA)-Clfa2 |
Bivalent fusion protein |
1581 |
527 |
33 |
mHla(H35L)-IsdB2 (mHIN2) |
Bivalent fusion protein |
1332 |
444 |
34 |
mHla(delta110-150)-IsdB1 |
Bivalent fusion protein |
1233 |
411 |
35 |
mHla(delta110-150)-IsdB2 |
Bivalent fusion protein |
1209 |
403 |
36 |
mHla(H35L)-FnbA1 |
Bivalent fusion protein |
2394 |
798 |
37 |
mHla(H35L)-FnbA2 |
Bivalent fusion protein |
2304 |
768 |
38 |
mHla(delta110-150)-FabA1 |
Bivalent fusion protein |
2271 |
757 |
39 |
mHla(delta110-150)-FnbA2 |
Bivalent fusion protein |
2181 |
727 |
40 |
SpA D(KA)- SpA D(KA) |
Bivalent fusion protein |
315 |
105 |
41 |
SpA D(KA)-FnbA1 |
Bivalent fusion protein |
1584 |
528 |
42 |
SpA D(KA)-FnbA2 |
Bivalent fusion protein |
1494 |
498 |
43 |
SpA D(KA)-SEB(RAA) |
Bivalent fusion protein |
951 |
317 |
44 |
LTB-SPAD(KA) |
Bivalent fusion protein |
474 |
158 |
45 |
SpA D(KA)-LTB |
Bivalent fusion protein |
474 |
158 |
46 |
SpA D(KA)-Clfa2 |
Bivalent fusion protein |
945 |
315 |
47 |
Clfa2-SPAD(KA) |
Bivalent fusion protein |
945 |
315 |
48 |
IsdB1-Clfa1 |
Bivalent fusion protein |
939 |
313 |
… |
… |
… |
… |
… |
Round 3
Reviewer 2 Report
The reply was comprehensive and acceptable.